# Co-Radiation of *Leptospira* and Tenrecidae (Afrotheria) on Madagascar

**DOI:** 10.3390/tropicalmed7080193

**Published:** 2022-08-18

**Authors:** Yann Gomard, Steven M. Goodman, Voahangy Soarimalala, Magali Turpin, Guenaëlle Lenclume, Marion Ah-Vane, Christopher D. Golden, Pablo Tortosa

**Affiliations:** 1Unité Mixte de Recherche Processus Infectieux en Milieu Insulaire Tropical (UMR PIMIT), Université de La Réunion, CNRS 9192, INSERM 1187, IRD 249, Plateforme Technologique CYROI, 97490 Sainte-Clotilde, France; 2Association Vahatra, BP 3972, Antananarivo 101, Madagascar; 3Field Museum of Natural History, Chicago, IL 60605, USA; 4Department of Nutrition, Harvard TH Chan School of Public Health, Boston, MA 02115, USA; 5Department of Environmental Health, Harvard TH Chan School of Public Health, Boston, MA 02115, USA

**Keywords:** microbial endemism, *Leptospira* *mayottensis*, leptospirosis, tenrecids, Madagascar, Mayotte

## Abstract

Leptospirosis is a bacterial zoonosis caused by pathogenic *Leptospira* that are maintained in the kidney lumen of infected animals acting as reservoirs and contaminating the environment via infected urine. The investigation of leptospirosis through a *One Health* framework has been stimulated by notable genetic diversity of pathogenic *Leptospira* combined with a high infection prevalence in certain animal reservoirs. Studies of Madagascar’s native mammal fauna have revealed a diversity of *Leptospira* with high levels of host-specificity. Native rodents, tenrecids, and bats shelter several distinct lineages and species of *Leptospira*, some of which have also been detected in acute human cases. Specifically, *L. mayottensis*, first discovered in humans on Mayotte, an island neighboring Madagascar, was subsequently identified in a few species of tenrecids on the latter island, which comprise an endemic family of small mammals. Distinct *L. mayottensis* lineages were identified in shrew tenrecs (*Microgale cowani* and *Nesogale dobsoni*) on Madagascar, and later in an introduced population of spiny tenrecs (*Tenrec ecaudatus*) on Mayotte. These findings suggest that *L. mayottensis* (i) has co-radiated with tenrecids on Madagascar, and (ii) has recently emerged in human populations on Mayotte following the introduction of *T. ecaudatus* from Madagascar. Hitherto, *L. mayottensis* has not been detected in spiny tenrecs on Madagascar. In the present study, we broaden the investigation of Malagasy tenrecids and test the emergence of *L. mayottensis* in humans as a result of the introduction of *T. ecaudatus* on Mayotte. We screened by PCR 55 tenrecid samples from Madagascar, including kidney tissues from 24 individual *T. ecaudatus*. We describe the presence of *L. mayottensis* in Malagasy *T. ecaudatus* in agreement with the aforementioned hypothesis, as well as in *M. thomasi*, a tenrecid species that has not been explored thus far for *Leptospira* carriage.

## 1. Introduction

Leptospirosis is a zoonotic disease that results annually in around 1 million human cases and nearly 60,000 deaths [1]. *Leptospira* bacteria, the pathogen responsible for the disease, are maintained in the lumen of the kidney tubules of animal reservoirs [2], which can chronically shed viable bacteria in their urine and contaminate the environment [3]. Although humans can be affected through direct contact with infected reservoirs, indirect transmission during outdoor activities in a contaminated environment is most frequent [4]. Infection leads to a wide range of symptoms ranging from mild flu-like syndromes to multi-organ failure causing death in 5–10% of the cases.

The genus *Leptospira* is currently composed of more than 60 taxa including saprophytic and pathogenic species [3,5,6,7,8]. Investigations carried out in different areas of the world through a *One Health* approach have shown distinct transmission chains composed of species or lineages and reservoirs that vary from one environmental setting to another [9,10,11,12,13]. Investigations carried out in the ecosystems of Madagascar and surrounding islands, hereafter referred to as the Malagasy Region, have provided new information on transmission chains on the different islands [14]. Indeed, on La Réunion and in the Seychelles, human leptospirosis is mostly caused by *Leptospira* that are broadly distributed and hence likely of introduced origin [11,12]. By contrast, Madagascar and Mayotte, a French-administrated island in the Comoros archipelago, shelter distinctly more diversified *Leptospira* assemblages, including species and lineages that are best considered endemic [15,16,17].

Among pathogenic *Leptospira* described and investigated in the Malagasy Region, *L. mayottensis*, the principal focus of the current study, warrants further characterization. These bacteria were first isolated from acute human leptospirosis cases on Mayotte and initially named *L. borgpetersenii* group B [9,18]. A thorough characterization of serological and genomic features of these isolates led the French Reference Centre on Spirochetes to elevate this bacterium to the rank of a new species, which was named *L. mayottensis* in reference to the geographic origin of the human isolates [19]. A comprehensive investigation of the Malagasy wild mammal fauna allowed identification of *Leptospira* samples imbedded in the genetic clade of *L. mayottensis* and shed by two endemic small mammal species, namely *Microgale cowani* and *Nesogale dobsoni* [20]. These two host species belong to the endemic family Tenrecidae, composed of omnivorous small mammals known to play an important role in *Leptospira* maintenance as reservoirs of two distinct species: *L. borgpetersenii* and *L. mayottensis* [17,20,21]. The origin of the Tenrecidae, a monophyletic group, is the result of a single colonization event originating from Africa that took place 30–56 million years ago, followed by an extraordinary radiation leading to the currently named nearly 40 extant species or confirmed candidate species [22,23]. These findings strongly suggest that *L. mayottensis* has co-radiated with tenrecid hosts on Madagascar.

It has been proposed that *L. mayottensis* was introduced to Mayotte from Madagascar [24]. This was supported by an investigation of animal reservoirs on Mayotte identifying *Tenrec ecaudatus*, a spiny Tenrec introduced from Madagascar for human consumption, as the local reservoir of *L. mayottensis*. However, the hypothesis that *T. ecaudatus* sheds *L. mayottensis* currently lacks definitive evidence for Malagasy populations of this species. In the present investigation, we screened *T. ecaudatus* specimens together with other tenrecid species sampled on Madagascar to broaden information on the presence of *L. mayottensis* in these animals, and to test the hypothesis of *L. mayottensis* being transported to Mayotte associated with the introduction of *T. ecaudatus*.

## 2. Materials and Methods

### 2.1. Biological Sample

All investigated shrew tenrecs (subfamily Oryzorictinae) were sampled in February 2016 in a forest neighboring the village of Anjozorobe, in the Central Highlands of Madagascar (see Figure 1). The samples included 31 specimens belonging to the following nine species: *Microgale taiva* (*n* = 15), *M. thomasi* (*n* = 3), *M. majori* (*n* = 3), *M. parvula* (*n* = 2), *M. soricoides* (*n* = 2), *M. cowani* (*n* = 1), *M. longicaudata* (*n* = 1), *M. fotsifotsy* (*n* = 1), and *Nesogale dobsoni* (*n* = 3). The spiny tenrec samples composed of *Tenrec ecaudatus* (subfamily Tenrecinae) included 24 specimens collected in villages adjacent to the Makira Natural Park in the Commune Antsirabe-Sahatany (Maroantsetra District) (Figure 1), an area with heavy human hunting pressure [25]. All samples in this region were collected from captured animals provided by local hunters to the research team. All specimens were captured, manipulated, and euthanized following guidelines accepted by the scientific community for the handling of wild mammals [26] and in strict accordance with permits issued by the national authorities of Madagascar. All kidney samples from the collected animals from both project areas were immediately stored in 70% ethanol until DNA extraction and molecular analyses.

### 2.2. Leptospira Detection and Sequencing

For DNA extraction, kidneys were first rinsed with water and subsequently immersed in 2 mL of sterile water overnight. Then, a thin transversal slice (approximately 0.5 mm thick) was cut in the central part of the kidney using a sterile scalpel, chopped into small pieces, and then submerged into lysis buffer provided in the DNeasy Blood and Tissue Kit (Qiagen, Hilden, Germany) used for DNA extraction. All subsequent extraction steps employed the manufacturer’s instructions. *Leptospira* detection was then carried out on 2 µL of eluted DNA using a probe-specific Real-Time Polymerase Chain Reaction system (RT-PCR) targeting a fragment of the 16S rRNA gene [28]. DNA templates leading to positive RT-PCR results were further subjected to end-point PCRs targeting the *secY* and *adk* loci of the MLST scheme#3, as previously described [29]. Amplicons were Sanger sequenced on both strands at GenoScreen (Lille, France) using the same PCR primers. The produced chromatograms were visually edited using Geneious software version 9.0.5 [30].

### 2.3. Statistical Analyses

Infection prevalence presented herein was compared to figures reported in a previous study (see Table 1 in [20]) using Fisher’s exact test, with a significance threshold set at *p* < 0.5.

### 2.4. Phylogeny

A phylogeny was constructed for the *secY* gene based on the bacterial sequences generated in the present study and previous *secY* sequences from other research in the Malagasy Region [9,15,17,20,24] (Appendix A), and different *Leptospira* species were used as ingroups and outgroups. The best model of sequence evolution was determined with jModelTest v.2.1.4 [31]. Phylogenetic reconstruction was performed with MrBayes v.3.2.3 [32]. The analysis consisted of two independent runs of four incrementally heated Metropolis Coupled Markov Chain Monte Carlo (MCMCMC) starting from a random tree. MCMCMC was run for 2 million generations with trees and associated model parameters sampled every 100 generations. The convergence level was validated by an average standard deviation of split frequencies inferior to 0.05. The initial 10% of trees for each run were discarded as burn-in and the consensus phylogeny along with posterior probabilities were obtained from the remaining trees. The resulting Bayesian phylogeny was visualized and annotated with FigTree v.1.4.2 [33].

## 3. Results and Discussion

The detection by RT-PCR indicates a global leptospiral infection rate of 7.3% (4/55) with bacteria detected in three out of the nine tested tenrecid species: *Microgale taiva* (one positive specimen), *M. thomasi* (two positive specimens), and *Tenrec ecaudatus* (one positive specimen). This overall prevalence is not significantly different from that reported in a previous study carried out in other areas of Madagascar, where 5.6% (12/213) of analyzed tenrecids tested positive for *Leptospira* [20]. The PCR protocols allowed leptospiral sequences to be obtained from the RT-PCR-positive *T. ecaudatus* (*secY*) and from one out of the two RT-PCR-positive *M. thomasi* (*secY* and *adk*). No bacterial sequence was obtained from the second RT-PCR-positive *M. thomasi* or from the RT-PCR-positive *M. taiva*. The three sequences were deposited in GenBank (Accession Numbers MT442041-MT442043).

We present in Figure 1 the Bayesian phylogeny obtained from the *secY* gene. Within this phylogeny, the bacterial sequences obtained from *T. ecaudatus* and *M. thomasi* fall in the *L. mayottensis* clade and form a well-supported subclade with a leptospiral sequence obtained from *Nesogale dobsoni*. This subclade is related to one subclade of *L. mayottensis* detected in humans and tenrecs from Mayotte. All previously reported *Leptospira* sequences from *Microgale* and *Nesogale* species are positioned within two distinct clades: *L. borgpetersenii* (*M. longicaudata, M. principula*, and *M. majori*) and *L. mayottensis* (*M. cowani* and *N. dobsoni*). Our results further support this topology with the detection of *L. mayottensis* in *M. thomasi* and Malagasy populations of *T. ecaudatus*.

The Tenrecidae are placental mammals grouped within a monophyletic family endemic to Madagascar and composed of nearly 40 species, including confirmed candidate species [22,23,34]. This highly diversified family is currently considered the result of a single colonization event originating from East Africa that took place between 30 and 56 million years ago, followed by speciation that resulted in an exceptional adaptive radiation [35,36]. Some tenrecids exhibit a number of biological features unique among mammals, such as the ability of hibernating without interbout arousal, partial heterothermy, or elementary echolocation [34,37].

The deep evolutionary history of the Tenrecidae also makes this family suitable for investigating the development of host–parasite interactions. For example, tenrecids host a diversity of Paramyxoviruses, some of which underwent host switches with introduced Muridae rodents [38]. Tenrecidae are known to be hosts of two species of pathogenic *Leptospira*, namely *L. borgpetersenii* and *L. mayottensis* [17,20,24]. While *L. mayottensis* has been identified in tenrecids (on Madagascar and Mayotte) and acute human cases (on Mayotte), a study on Madagascar reported the presence of *L. mayottensis* in introduced *Rattus rattus*, but only as co-infections with other *Leptospira* species [39]. The strong host-specificity of *L. mayottensis* towards tenrecids was recently tested through experimental infection in which *L. mayottensis* isolated from *T. ecaudatus* failed to colonize the kidneys of *R. norvegicus* [40]. The present study was carried out to (i) further explore the diversity of *L. mayottensis* sheltered by tenrecids and (ii) confirm a previous hypothesis that proposed *L. mayottensis* arrived on Mayotte with the introduction of *T. ecaudatus* for human consumption.

Analyzed samples confirmed tenrecids as being a reservoir of *L. mayottensis* and added *M. thomasi* to the list of animal reservoirs of this pathogenic bacteria. Of particular importance, we report the first characterization of *L. mayottensis* from *T. ecaudatus* on Madagascar. Together with previous data reported on Mayotte [24], the present work supports the introduction of this mammal species to Mayotte being associated with the emergence of a zoonotic human pathogen, *L. mayottensis* on that island. *Tenrec ecaudatus* has also been introduced to other islands in the Malagasy Region with the purpose of providing bush meat, most notably La Réunion, Mauritius, Mahé (Seychelles), and other islands in the Comoros archipelago, but to our knowledge *L. mayottensis* has not been isolated in these non-native *T. ecaudatus* populations or reported in local human inhabitants. The *L. mayottensis* infection prevalence measured in *T. ecaudatus* (4.2%, *n* = 24) is significantly lower than previously reported on Mayotte (27%, *n* = 37; see [24]), while no positive animals have been reported on La Réunion in two independent studies [11,41]. This pattern might be the result of different origins of the *T. ecaudatus* populations introduced to western Indian Ocean islands. On Madagascar, *T. ecaudatus* is found in a range of different forest types and it would be interesting to document the possible phylogeographic structure of these populations and then try to determine the most plausible geographic origin of the populations that have been introduced to other islands in the Malagasy Region. Alternatively, environmental conditions might be more conducive to *L. mayottensis* transmission among *T. ecaudatus* on Mayotte Island than on Madagascar or La Réunion, a hypothesis challenging to test as it requires comprehensive information on *L. mayottensis* biology, including environmental survival of these bacteria in the different geological and climatic contexts.

Data presented herein support *L. mayottensis* being a zoonotic pathogen originating from Madagascar, although we emphasize that overall infection prevalence is low and, hence, preclude any definitive conclusion. However, in addition to the reports of *L. mayottensis* in tenrecs from Madagascar and neighboring Mayotte, it is important to mention that *L. mayottensis* has not been reported outside of western Indian Ocean islands, an area that has considerable species diversity of small mammals including tenrecs and native rodents [42]. We therefore propose that long-term co-radiation processes between Malagasy endemic small mammals and their hosted infectious agents have led to the emergence of endemic microorganisms with zoonotic potential, such as *L. mayottensis*. It has been hypothesized nearly a century ago that the extreme abundance and unbounded dispersal capacities of microorganisms limit endemism, with the exception of some extreme environments, and that biogeographical patterns result from contemporary selective pressures rather than from limited dispersal capacity. This dogma, often referred to as the Baas Becking hypothesis—“*everything is everywhere but the environment selects*” [43]—has been increasingly challenged, but microbial biogeography is still in its infancy [44,45]. The present study supports that host-specificity needs to be considered as a driver of microbial endemism: the dispersal capacities of host-specific microbes are indeed limited by that of their hosts. In other words, when considering host–parasite pairs, the dispersal capacities of hosts drive the biogeographical patterns of their associated microorganisms and may, in the case of strong host–parasite specificity, lead to microbial endemism.

## Figures and Tables

**Figure 1 tropicalmed-07-00193-f001:**
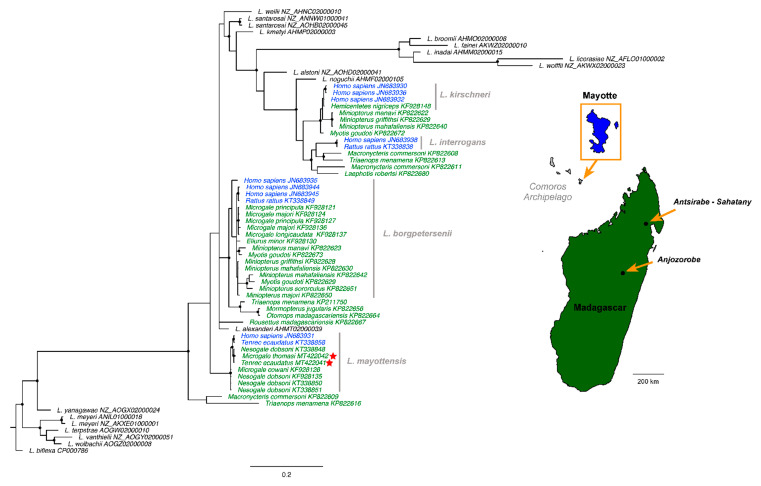
Geographical context and Bayesian phylogenetic tree of *Leptospira* species from Mayotte (blue) and Madagascar (green) based on *secY* gene (482 bp). Sequences in black correspond to *Leptospira* species used as ingroups and outgroup (*L. biflexa*). The accession number is indicated for each sequence. The analysis was conducted under the HKY + I + G substitution model. Black circles at the nodes indicate posterior probabilities superior or equal to 0.90. The red stars indicate new sequences generated in the present study and were obtained from two regions on Madagascar: Anjozorobe and Makira (Commune Antsirabe-Sahatany). The map was realized using worldHires function in mapdata package [27] under the R software version 4.1.1.

## Data Availability

The produced sequences were deposited in GenBank under the accession numbers MT442041, MT442042, and MT442043.

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
