# Peer review of "Co-Radiation of Leptospira and Tenrecidae (Afrotheria) on Madagascar"

_tropicalmed, 2022, doi:10.3390/tropicalmed7080193_

Round 1

Reviewer 1 Report

This paper by Gomard et al decribes the identification of L. mayottensis in small mammals in Madagascar by sequencing. The study is scientifically sound and explores an interesting hypothesis regarding the co-speciacion of pathogenic Leptospira and its reservoirs. However (unfortunately!), the rate of infection of the mammals captured is very small and so, in my opinion, the conclusions should be toned down and the limitations be candidly acknowledged.

SPECIFIC COMMENTS:

- In the phylogeny trees, only a few pathogenic species were included. Given that the two sequences obtained cluster well with L. mayottensis makes this comment perhaps less relevant, but I would suggest that in in future studies all the pathogenic species (P1 and P2 groups) are included.

- Were any efforts made to obtain good sequences from the other two positive samples? i.e. nPCR with G1/G2 fragments or any other nested/semi-nested procedure to improve signal?

- The authors should elaborate on the discussion on why the positive rate was so low (7.3%). Was this expected? What are the positive rates in other studies with similar mammals? Given the low positive rate in T. ecaudatus, could this indicate that it is perhaps not a chronic carrier of Leptospira, but rather a susceptible animal? What are potential future studies that can address this? How this low positive rate can affect the dynamics of transmission between animal reservoirs and to humans?

- As explained above, the extremely small number of positive samples and even smaller sequences obtained from them, limits greatly the ability to conclude much about co-radiation or origins of L. mayottensis in Madagascar/Mayotte. Essentially, the entire hypothesis is based on a single sequence found in one animal. Although I value the great effort done by the authors, I believe the discussion section should be rewritten to tone down the conclusions about co-radiation (i.e. the first sentence of the concluding paragraph says that the data presented strongly support...; this is an overstatement from my point of view) and clearly explain that the co-radiation hypothesis is really speculative at this point. Also, please include a paragraph acknowledging in depth the limitations of the small amount of data that this study is based on).

Reviewer 2 Report

This is a survey aiming to describe the presence of L. mayottensis in Malagasy T. ecaudatus and M. thomasi. The subject is interesting and the results are important. Please find below some suggestions to improve the paper:

1. ABSTRACT: the abstract must include a short (two or three lines) introduction, objective, methodology, results and conclusion. Of the 18 lines of abstract 15 were used for introduction. 

2. Page 2, line 83, and Page 4, line 172: what means "My ago" ?

3. Page 5, lines 201-202: "In conclusion, the data presented herein also strongly support that L. mayottensis is an endemic zoonotic pathogen to Madagascar". The data of the survey do not support this statement. Please delete or modify.

Reviewer 3 Report

I find that your work is well written and even if the contents are complex, the results are clearly described.

You have done a lot of work, both as a search for samples, as a laboratory analysis and as a genetic analysis.

For me the work is worth publishing.

Round 2

Reviewer 1 Report

Thanks for thoroughly addressing all my comments/suggestions. I hope they contributed to making this study stronger.